# Debugging Tabular Log as Dynamic Graphs

## Abstract

Tabular log abstracts objects and events in the real-world system and reports their updates to reflect the change of the system, where one can detect real-world inconsistencies efficiently by debugging corresponding log entries. However, recent advances in processing text-enriched tabular log data overly depend on large language models (LLMs) and other heavy-load models, thus suffering from limited flexibility and scalability. This paper proposes a new framework, GraphLogDebugger, to debug tabular log based on dynamic graphs. By constructing heterogeneous nodes for objects and events and connecting node-wise edges, the framework recovers the system behind the tabular log as an evolving dynamic graph. With the help of our dynamic graph modeling, a simple dynamic Graph Neural Network (GNN) is representative enough to outperform LLMs in debugging tabular log, which is validated by experimental results on real-world log datasets of computer systems and academic papers.

## 1 Introduction

Tabular log data plays a crucial role in representing and tracking the state and evolution of real-world systems. These logs are structured as rows of log entries, each capturing an event involving certain objects and their attributes at a specific time point. Common examples include system logs recording computing services (Zhu et al., 2023a), research logs tracking scientific publication activities (Clement et al., 2019), and interaction logs from multi-agent systems powered by large language models (LLMs) (Zhang et al., 2025b). Debugging of tabular logs is essential: it allows practitioners to detect anomalies in the original systems through efficient inspection of associated log records.

Log anomaly detection (He et al., 2016) has therefore been a long-standing research field in different niche areas, where data distributions are invariant or have little change. Existing frameworks (Du et al., 2017; Meng et al., 2019; Zhang et al., 2019; Pei et al., 2020; Guo et al., 2021; Chen & Tsourakakis, 2022) benefit from manually defined data structures or templates for log parsing which are often tailored to certain domains and thus yield absolute success in specific areas like computer system log or financial event log. However, due to this domain-specific principle, designing a general-purpose log debugger always remains challenging.

Efforts to overcome this challenge have led to two main lines of work, as shown in Figure 1. One stream focuses on graph modeling of the log data (Cheng et al., 2020; Zehra et al., 2021; Pang et al., 2025), where information in tabular log is gathered in a unified data structure: the graph, such as constructing knowledge graphs or text-rich dynamic graphs for computer system log (Sui et al., 2023; Li et al., 2023). Although these methods are both efficient and powerful, many of them lack flexibility: they still customize static graph structures for certain domains. Another stream explores LLM-based solutions, such as LLM prompting (Yu et al., 2023; Qi et al., 2023; Park, 2024) or retrieval-augmented generation (RAG) (Pan et al., 2024; Zhang et al., 2025a; Wang et al., 2025) pipelines. While these methods demonstrate general capabilities in text-based reasoning, thus showing potential of generalization, they often come with significant drawbacks: high computational costs, slow inference, and difficulty scaling to long log streams or resource-constrained settings.

Inspired by the idea to unify multimodal information in dynamic graphs (Feng et al., 2025), we propose **GraphLogDebugger**, a general and efficient framework for debugging tabular logs through dynamic graph modeling. Our core idea is to interpret tabular log entries as the evolving state of a hidden system, which can be reconstructed as a dynamic heterogeneous graph. We treat objects and events as different types of nodes with text embeddings empowered by modern language embed-

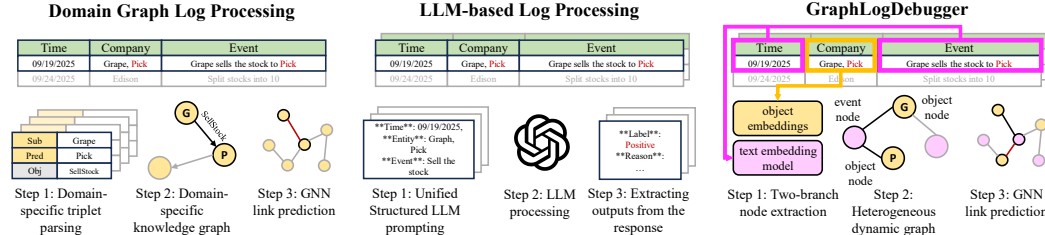

Figure 1: **Comparing GraphLogDebugger with two existing lines of works**. Processing log with domain-specific graphs requires custom text parsing, which lacks flexibility. LLM-based log processing overcomes this shortcoming by the general comprehension skills of LLMs, but suffers from poor efficiency. GraphLogDebugger combines the advantages of graph representation and those of LLMs and balances well generalizability and scalability.

ding models, and use the tabular structure to generate time-stamped connections between them. As new log entries arrive, they incrementally update the dynamic graph, capturing both structural and temporal dependencies. This formulation allows us to apply a lightweight dynamic Graph Neural Network (GNN) to perform online anomaly detection by evaluating the likelihood of new connections. Our approach avoids reliance on heavy LLMs while still capturing rich semantic and relational information in the data. Experimental results on real-world datasets from computer system logs and scientific publication logs validate the effectiveness of our approach. Despite its simplicity, our dynamic GNN framework outperforms LLM-based baselines in both accuracy and efficiency, demonstrating that dynamic graph modeling is a highly expressive yet lightweight alternative. Our contributions can be summarized as follows:

- We introduce a novel view of tabular logs as dynamic heterogeneous graphs, bridging the gap between structured attributes and semantic reasoning, and redefine the framework of online log anomaly detection, where object-event connections in each incoming log are evaluated through link prediction on the evolving graph.

- We propose a lightweight GNN-based debugger that can efficiently and accurately detect anomalies without using LLMs, and validate its performance on real-world datasets with diverse modalities.

## 2 RELATED WORKS

**Tabular Log Processing.** Many real-world logs include structured, time-stamped tabular attributes alongside annotated text fields. Examples come from financial prices paired with event series (Tetlock, 2007; Ruiz et al., 2012; Dong et al., 2024), scientific publication metadata (Clement et al., 2019; Kinney et al., 2023), healthcare records (Johnson et al., 2023), computer system logs (Zhu et al., 2023a), and multi-agent system reports (Zhang et al., 2025b). A key challenge in processing tabular logs with machine learning lies in capturing multi-attribute correlations while maintaining comprehension of their semantics (Wu et al., 2025). One common approach integrates main attributes recognized by human priors into structured data (Yang et al., 2018; Zhao & Feng, 2022; Koval et al., 2024), and then subsequently augments the representation by retrieval (Kurisinkel et al., 2024; Xiao et al., 2025). This modeling achieves good performance in domain-specific data, but lacks flexibility and generalizability for adaptation to other fields (Gardner et al., 2024).

An emerging alternative leverages Large Language Models (LLMs) (Brown et al., 2020), which have demonstrated strong generalizability in understanding, predicting, and generating tabular data (Liu et al., 2023; Zhang et al., 2024b; Fang et al., 2024; Wang et al., 2024b). By parsing diverse logs into a unified format with LLMs (Zhong et al., 2024), these models can be applied to downstream tasks that require reasoning capabilities, such as predicting stock prices (Yu et al., 2023), electricity demand (Wang et al., 2024a), and future events (Shi et al., 2023; Ye et al., 2024). However, LLM-based approaches often suffer from high overheads, complex deployment, and limited throughput. There remains a strong need for lighter-weight alternatives with comparable performance.

**Dynamic Graphs.** Graph Neural Networks (GNNs) have become a foundational paradigm for learning on graph-structured data (Kipf, 2016; Hamilton et al., 2017; Gilmer et al., 2017). Static

GNN models have benefited from advances in message passing (Battaglia et al., 2018), architectural depth (Li et al., 2021; Dwivedi et al., 2020), and inductive scalability (Hamilton et al., 2017). However, many real-world systems are dynamic, motivating models that capture both structural and temporal dependencies. Early approaches used recurrent layers or time-aware embeddings (Li et al., 2017; Seo et al., 2018) to extend static GNNs to dynamic settings (Pareja et al., 2020; Sankar et al., 2020; Kumar et al., 2019). Recent methods have embraced memory modules (Rossi et al., 2020) and temporal encoding (Xu et al., 2020) for finer-grained modeling of time-stamped interactions. Building on this trajectory, ROLAND (You et al., 2022) offers a framework that adapts static GNNs to dynamic graphs via hierarchical state propagation and live-update evaluation, which inspires new advances in benchmarks (Longa et al., 2023; Huang et al., 2023; Zhang et al., 2024a), architectures (Zhu et al., 2023b), explainability (Chen & Ying, 2023), and robustness (Zhang et al., 2023b).

**Log Anomaly Detection.** Log-based anomaly detection has long been a critical task for system reliability, and early neural approaches typically rely on sequence modeling via LSTMs (Du et al., 2017), CNNs (Lu et al., 2018), and autoencoders (Zhang et al., 2021; Castillo et al., 2022; Zhang et al., 2023a). Others incorporate adversarial training (Duan et al., 2021; He et al., 2023), or temporal networks (Zhang et al., 2019; Yang et al., 2021). More recently, pretrained language models have been adopted for log anomaly detection, either via fine-tuning (Guo et al., 2021; Lee et al., 2023) or prompt-based pipelines (Qi et al., 2023; Liu et al., 2024). Retrieval-augmented (No et al., 2024; Pan et al., 2024; Zhang et al., 2025a) methods have further pushed semantic understanding in LLM-based methods. As mentioned, while machine learning-based methods are highly domain-specific, LLM-based methods show some generalizability at a high cost.

One potential solution towards general and scalable methods for log debugging is to introduce dynamic graphs, where tabular log is considered as an evolving system and maintained in a dynamic graph. Early exploration makes use of knowledge graphs (Hogan et al., 2021) with domain specific parsing to generate triplets (Cheng et al., 2020; Zehra et al., 2021; Sui et al., 2023). Recent advances adopt dynamic graphs with text-rich nodes to represent tabular log (Li et al., 2023; Pang et al., 2025). Nevertheless, these works are either domain specific or LLM-based, yet not escaping from the dilemma between generalizability and scalability.

## 3 PRELIMINARIES

Tabular log is the data modality used to report the update of real-world systems from the perspective of states and relations. It can be formally defined by a time series $X = \{x_0, x_1, x_2, ..., x_{N-1}\}$ annotated by a timestamps sequence $t_0 < t_1 < t_2 < ... < t_{N-1}$, where each of $x_n$ is a log entry that contains different attributes $x_n^m$ in the table: $x_n = \{x_n^0, x_n^1, x_n^2, ..., x_n^{M-1}\}$. Summarizing the general case of tabular log data in finance (Dong et al., 2024), healthcare (Johnson et al., 2023), academics (Clement et al., 2019), and other systems (Zhu et al., 2023a), we can separate attributes in the tabular log into three types:

- **Object:** Attributes that represent stand-alone objects in the tabular log, such as companies in the financial news log and cities in the medical record log.

- **Event:** Attributes that describe an event with text, for example, news content in the financial news log and record content in the medical record log. These attributes are usually the center of log entries, where other attributes supplement details and involved objects of the event. Without loss of generality, one log entry only has one Event attribute, because we could merge the text sections of different event attributes into one.

- **Feature:** Attributes that describe features related to the event or objects. For instance, the age is a feature of the patient object in the medical record log. Timestamp $t_n$ is a special type of feature that provides the details about the time of the event.

In practice, we find that Objects and Features are mutually convertible. For example, the address of the company could be either an independent object or a feature of the company object in the financial log. Hence, the arrangement of Objects and Features is a hyperparameter that needs pre-definition.

While tabular log abstracts the change of real-world systems, it is expected that we could detect inconsistencies of the system from the corresponding tabular log. Based on the above categorization,

we could then define three types of anomalies and corresponding anomaly detection tasks in the tabular log. Give a log entry $x_n = \{x_n^0, x_n^1, x_n^2, ..., x_n^{M-1}\}$ in the tabular log $X$:

- **Object anomaly:** Let $\{o_n^0, o_n^1, o_n^2, ..., o_n^{P-1}\}(P < M)$ be the object set. We have label $y_n = \{y_n^0, y_n^1, y_n^2, ..., y_n^{P-1}\}$, where $y_n^m = 0$ means that $o_n^p$ is a normal object and $y_n^m = 1$ means that $o_n^p$ is an abnormal object for the log entry $x_n$.

- **Event anomaly:** Let $s_n$ be the event. We have a label $y_n$, where $y_n = 0$ means that $s_n$ is a normal event and $y_n = 1$ means that $s_n$ is an abnormal event for the log entry $x_n$.

- **Feature anomaly:** Let $\{f_n^0, f_n^1, f_n^2, ..., f_n^{Q-1}\}(Q < M)$ be the feature set. We have label $y_n = \{y_n^0, y_n^1, y_n^2, ..., y_n^{Q-1}\}$, where $y_n^q = 0$ means that $o_n^q$ is a normal feature and $y_n^q = 1$ means that $o_n^q$ is an abnormal feature for the log entry $x_n$.

Considering the tabular log $X$ as an online system where new log entries come dynamically in time order, we could then define the anomaly detection in tabular log as an online anomaly detection task:

**Definition 3.1** (**Online Anomaly Detection of Tabular Log**). Given an online system that dynamically produces log entry $x_n$, online anomaly detection for tabular log predicts its anomaly label $y_n$ based on historical log entries $X_n = \{x_0, x_1, x_2, ..., x_{n-1}\}$

Notably, object anomaly and feature anomaly are isomorphic. Considering the fact that objects and features are convertible, the rest of this paper only studies object anomaly and event anomaly. Table 3 summarizes all used variables.

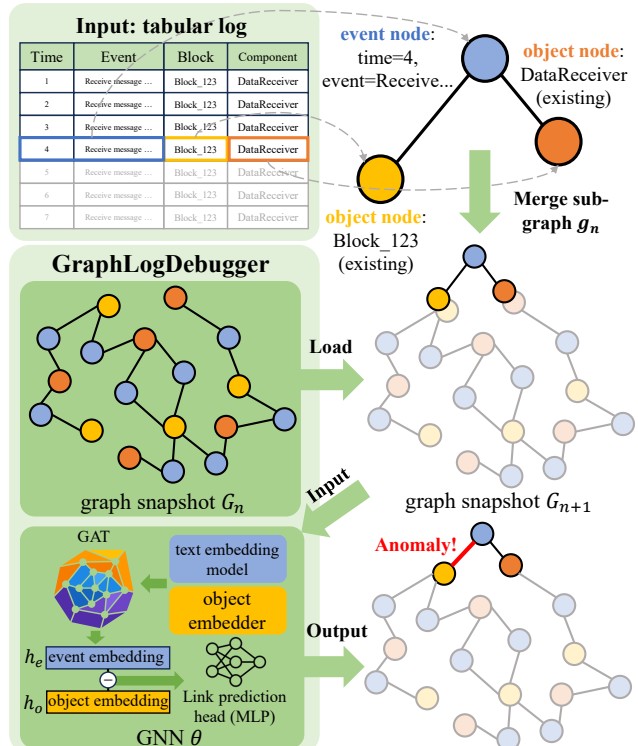

Figure 2: **GraphLogDebugger framework**. The framework checkpoints the GNN $\theta$ and the dynamic graph snapshot $G_n$. When a new log entry emerges, we first extract a sub-graph $g_n$ and use it to update the dynamic graph. Then, we predict the links introduced by $g_n$ in the dynamic graph by GNN $\theta$, whose results indicate the anomaly.

## 4 GRAPHLOGDEBUGGER

We first integrate tabular log to a heterogeneous dynamic graph (Section 4.1). Then, we reformulate online anomaly detection of tabular log as dynamic graph anomaly detection(Section 4.2). Finally, we apply a dynamic GNN to debug the tabular log (Section 4.3).

### 4.1 INTEGRATING ONLINE TABULAR LOG TO DYNAMIC GRAPHS

Objects and events in the same log entry are naturally connected in the tabular log, from which we could construct graphs. To this end, we first define the graph structure within one log entry. As shown in Figure 2 (upper section), we build nodes $v$ for both objects and the unique event in the new log entry. Each event and all its objects are connected by an edge $e$. This yields a sub-graph $g_n$ for each log entry $x_n$.

Figure 2 (middle section) illustrates the composition of a dynamic graph $\mathcal{G}$ that stores the information of all historical log entries. This dynamic graph gathers all sub-graphs of log entries. We merge identical object nodes so that these sub-graphs are connected. Note that every event node should be unique. Every time a new log entry $x_n$ emerges, we construct a sub-graph $g_n$ accordingly and merge it into the dynamic graph $\mathcal{G}$. We denote the snapshot of $\mathcal{G}$ at time point $t_n$ by $G_n$.

**Algorithm 1:** GraphLogDebugger: Online training for dynamic-graph anomaly detection

**Input** : Training log $X_{\text{train}} = \{x_{t_0}, \ldots, x_{t_{K-1}}\}$; GNN $\theta$; text embedding model $\mathcal{F}$; negative sampling ratio $\rho$; threshold $\tau$;

**Output:** Trained parameters $\theta^{\star}$, dynamic graph snapshot $G_{t_K}$

1 Initialize $\mathcal{G} : \mathcal{V} = \emptyset, \mathcal{E} = \emptyset$

2 **for** $k = 0, \ldots, K - 1$ **do**

    // Integrate the incoming log entry

3    Build sub-graph $g_k$: creating nodes for object set $\mathcal{V}_k^o$ and event set $\mathcal{V}_k^e$ in $x_{t_k}$ and connecting object-event pairs in $x_{t_k}$ with edges

4    $\mathcal{G} \leftarrow (\mathcal{V} \cup \mathcal{V}_k^o, \mathcal{E})$

5    Positive set $\mathcal{E}_k^+ \leftarrow$ object-event links in $g_k$.

6    Negatives $\mathcal{E}_k^-$ by drawing $\rho \cdot |\mathcal{E}_k^+|$ non-existent object–event pairs in $\mathcal{G}$.

    // Embed nodes

7    Compute object embeddings $h_o$ with GNN $\theta$: $h_o = f_\theta(\mathcal{G})$

8    Compute new event embeddings $h_e$ with text embedding model $\mathcal{F}$: $h_e = \mathcal{F}(\mathcal{V}_k^e)$

    // Predict links & Compute the loss

9    For each pair $(o, e) \in \mathcal{E}_k^+ \cup \mathcal{E}_k^-$, compute score $s_{o,e} = \sigma(\text{MLP}(\text{reduce}(h_o, h_e)))$.

10    Compute balanced BCE loss $\mathcal{L}_k$ on labels (1 for $\mathcal{E}_k^+$, 0 for $\mathcal{E}_k^-$) and update $\theta \leftarrow \theta - \eta \nabla_\theta \mathcal{L}_k$.

    // Updating the dynamic graph

11    $\mathcal{G} \leftarrow (\mathcal{V} \cup \mathcal{V}_k^e, \mathcal{E})$

12 Repeat Step 1-10 for epochs

13 **return** $\theta$

---

**Algorithm 2:** GraphLogDebugger: Online evaluation for dynamic-graph anomaly detection

**Input** : Test log $X_{\text{test}} = \{x_{t_K}, \ldots, x_{t_{N-1}}\}$; trained GNN $\theta^{\star}$; last snapshot $G_{t_K}$; text embedding model $\mathcal{F}$; threshold $\tau$

**Output:** Per-time link predictions $\{\mathcal{R}_{t_n}\}_{n=K}^{N-1}$; updated snapshot $G_{t_N}$

1 Initialize $\mathcal{G} \leftarrow G_{t_K}$ and load $\theta^{\star}$

2 **for** $n = K, \ldots, N - 1$ **do**

    // Integrate the incoming log entry

3    Build sub-graph $g_n$ from $x_n$ with object set $\mathcal{V}_n^o$, new event set $\mathcal{V}_n^e$, and observed links $\mathcal{E}_n^+$

4    $\mathcal{G} \leftarrow (\mathcal{V} \cup \mathcal{V}_k^o, \mathcal{E})$

    // Embed nodes

5    Compute object embeddings on current snapshot: $h_o = f_{\theta^{\star}}(\mathcal{G})$

6    Compute embeddings for new events (time-aware): $h_e = \mathcal{F}(\mathcal{V}_n^e)$

    // Predict links

7    For each $(o, e) \in \mathcal{E}_n^+$, compute $s_{o,e} = \sigma(\text{MLP}(\text{reduce}(h_o, h_e)))$ and set $\hat{\ell}_{o,e} = \mathbb{1}[s_{o,e} \geq \tau]$

8    **Link prediction results:** $\mathcal{R}_{t_n} \leftarrow \{(o, e, s_{o,e}, \hat{\ell}_{o,e}) \mid (o, e) \in \mathcal{E}_n^+\}$

    // Update the dynamic graph

9    Accepted links $\hat{\mathcal{E}}_n^+ = \{(o, e) \in \mathcal{E}_n^+ \mid \hat{\ell}_{o,e} = 1\}$

10    Accepted new events $\hat{\mathcal{V}}_n^e = \{e \in \mathcal{V}_n^e \mid \exists o : (o, e) \in \hat{\mathcal{E}}_n^+\}$.

11    $\mathcal{G} \leftarrow (\mathcal{V} \cup \hat{\mathcal{V}}_n^e, \mathcal{E} \cup \hat{\mathcal{E}}_n^+)$

12 **return** $\{\mathcal{R}_{t_n}\}_{n=K}^{N-1}$ and $G_{t_N} = \mathcal{G}$

---

## 4.2 DEBUGGING TABULAR LOG GRAPHS AS DYNAMIC GRAPHS

Following the above integration, we transfer the online anomaly detection of tabular log defined in Section 3 into an anomaly detection problem of dynamic graph $\mathcal{G}$ (Ekle & Eberle, 2024):

- **Object anomaly detection:** The object anomaly occurs when the edge $e$ between an object node and an event node is abnormal in the latest sub-graph $g_n$. This anomaly could then be detected by link prediction in the dynamic graph $\mathcal{G}$ (You et al., 2022), where a GNN is applied to predict the likelihood of $e$. If the likelihood exceeds a threshold, we consider the edge as normal. Otherwise, we consider the edge as an anomaly.

- **Event anomaly detection:** The event anomaly occurs when an abnormal event $s$ is placed in the wrong entry in the latest sub-graph $g_n$. This means that all edges between this event are anomalies. We can therefore apply link prediction to all edges in the sub-graph and threshold the overall predicted likelihoods to determine the anomaly label of the event.

In a nutshell, the goal of our anomaly detection in Section 3 is equivalent to predicting the likelihood of all edges in the latest log entry sub-graph $g_n$, based on the dynamic graph snapshot $G_n$ that stores all historical log entries of tabular log $X_n$. We finally transfer the online anomaly detection of tabular log into an anomaly detection problem in dynamic graphs, with notations omitted to Table 3:

**Definition 4.1** (**Online Anomaly Detection of Tabular Log (Dynamic Graph)**)**.** Given the snapshot $G_n$ of a dynamic graph $\mathcal{G} = \{G_n\}_{n=0}^{N-1}$ and the new coming sub-graph $g_n = G_{n+1} \setminus G_n$, the goal is to predict the label $y_n$ of links in $g_n$.

Table 1: **Statistics and details of the four datasets for tabular log debugging.**

| Dataset | Domain | #Entries | #Objects | Event Attr. | Obj Attr. | Anomaly Type |
|---------|--------|----------|----------|-------------|-----------|--------------|
| Arxiv | Sci. Pub. | 20,000 | 17316 | title | authors | Event/Object |
| HDFS | System Log | 20,000 | 2150 | Content | Component,EventId,BlockId | Object |
| Analyst | Finance | 20,000 | 3901 | headline | publisher | Event |
| Landslide | Geology | 20,000 | 8565 | description | title, category,trigger,country | Object |

### 4.3 DESIGNING THE GNN FOR DYNAMIC GRAPH ANOMALY DETECTION

Figure 2 (bottom section) demonstrates the basic process of using our GNN to predict link anomaly labels. For details, our GNN $\theta$ takes the dynamic graph snapshot $G_n$ and the incoming sub-graph $g_n$ as inputs and predicts the likelihood of all links in $g_n$. The GNN consists of three parts: the node embedder, the GNN backbone, and the prediction head. First, the node embedder offers heterogeneous embeddings for all objects in $G_n$ and $g_n$ and event nodes in $G_n$. We exclude new events in $g_n$ because we do not expect the outputs of the model will be interfered with by the graph structure in $g_n$. We assign a unique learnable embedding for each object, and use a pre-trained text embedding model to embed existing events. We also concatenate a time embedding to the event embedding based on the coming time $t_n$ for event $s$. Our GNN backbone is adapted from the graph attention network (GAT) (Veličković et al., 2017), where we use two separate MLPs to map objects and events to the same space and apply GAT layers for message passing. The prediction head predicts the link between all object-event pairs in $g_n$. We first compare object embeddings after GAT layers and event text embeddings by reduction. We then pass the result to an MLP with Sigmoid activation to get the likelihood. We omit more details in the design space of the model in Appendix A.3.

Our GNN is trained under the setting of unsupervised anomaly detection (Pang et al., 2021): We separate the dataset into a training split and a test split by chronological order. In the training stage, all links in $g_n$ are normal, and we provide negative examples for training by randomly sampling object-event pairs that are not connected. We append these fraud links to the ground-truth links to balance the label distribution and use them to train the GNN. After backpropagation, we finally update the dynamic graph snapshot $G_n$ with subgraph $g_n$. Alg 1 summarizes the training algorithm.

In the test stage, we first resume the GNN as well as the latest dynamic graph snapshot $G_n$. This achieves the warm start of our debugger system. Then, we construct sub-graphs from the coming log entries and take them as parts of the evolving dynamic graph $\mathcal{G}$ that we succeed from the pre-trained GNN. The evaluation process is summarized in Alg 2.

## 5 EXPERIMENTS

### 5.1 EXPERIMENTAL SETTINGS

**Datasets.** Our work provides a general framework of debugging different types of tabular log under the online setting. To validate this point, we span our experiments over datasets covering four different fields: (1) **Arxiv**: Tabular log recording the timestamps (from 2007-2025), the title and the authors of machine learning papers from the Arxiv (Clement et al., 2019) API; (2) **HDFS**: system log of Hadoop Distributed File System designed to run on commodity hardware (Xu et al., 2009; Zhu et al., 2023a), including the event content together with the objects related to the event; (3) **Analyst**: the commentary records on the finance by analysts, including title, author, and other features of posts [1]; (4) **Landslide**: event catalog reporting the global landslide [2]. These datasets contain both text-rich attributes and categorical attributes with diversified semantics, thus being challenging to process in one framework efficiently. Table 1 demonstrates the basic statistics of our four datasets.

We limit the maximum length of all tabular logs to 20,000 by slicing the original datasets. This is because LLM-based baseline methods are costly and not scalable, as discussed in the introduction. To ensure randomness, we randomly pick slices with a length of 20,000 from the whole sliced dataset. For datasets with multiple object attributes, we evaluate object anomaly detection, while for those with only one object attribute, we evaluate event anomaly detection, where event and object anomaly detection are equivalent. We summarize the basic setting of our four datasets in Table 1.

---

[1] www.kaggle.com/datasets/miguelaenlle/massive-stock-news-analysis-db-for-nlpbacktests
[2] https://catalog.data.gov/dataset/global-landslide-catalog-export

Table 2: **Our proposed GraphLogDebugger outperforms representative baselines on detection effectiveness and efficiency across diverse methods and datasets.** Higher is better for detection effectiveness; lower GFLOPs and higher Throughput are preferred for efficiency. "*" suggests some baselines always predict non-anomaly cases, leading to a 0 prediction, recall, and F1 score.

| | Dataset: **Arxiv** Task: **Event Anomaly** | | | | | |
| Method | Detection Effectiveness | | | | Efficiency | |
| | Acc. | Prec. | Recall | F1 | GFLOPs | Throughput (it/s) |
|---|---|---|---|---|---|---|
| MLP | $0.570 \pm 0.205$ | $0.556 \pm 0.189$ | $0.893 \pm 0.331$ | $0.676 \pm 0.028$ | 11.35 | $825.0 \pm 1429.0$ |
| RAG (Llama3-70b,$k$=5) | $0.408 \pm 0.038$ | $0.426 \pm 0.026$ | $0.527 \pm 0.029$ | $0.471 \pm 0.019$ | $\sim 10^5$ | $0.204 \pm 0.009$ |
| RAG (GPT-oss-20b,$k$=5) | $0.770 \pm 0.106$ | $0.771 \pm 0.157$ | $0.773 \pm 0.014$ | $0.772 \pm 0.085$ | $\sim 10^4$ | $0.145 \pm 0.018$ |
| RAG (Llama3-70b,$k$=10) | $0.377 \pm 0.014$ | $0.400 \pm 0.004$ | $0.493 \pm 0.038$ | $0.442 \pm 0.014$ | $\sim 10^5$ | $0.204 \pm 0.015$ |
| RAG (GPT-oss-20b,$k$=10) | $0.803 \pm 0.090$ | $0.798 \pm 0.099$ | $0.813 \pm 0.087$ | $0.805 \pm 0.087$ | $\sim 10^4$ | $0.149 \pm 0.014$ |
| GraphLogDebugger (Ours) | $\mathbf{0.957} \pm 0.040$ | $\mathbf{0.920} \pm 0.069$ | $\mathbf{1.000} \pm 0.000$ | $\mathbf{0.959} \pm 0.037$ | 40.39 | $627.662 \pm 7.623$ |

| | Dataset: **Arxiv** Task: **Object Anomaly** | | | | | |
| Method | Detection Effectiveness | | | | Efficiency | |
| | Acc. | Prec. | Recall | F1 | GFLOPs | Throughput (it/s) |
|---|---|---|---|---|---|---|
| MLP | $0.570 \pm 0.000$ | $0.538 \pm 0.000$ | $0.990 \pm 0.000$ | $0.697 \pm 0.000$ | 11.35 | $552.0 \pm 167.0$ |
| RAG (Llama3-70b,$k$=5) | $0.455 \pm 0.033$ | $0.468 \pm 0.026$ | $0.667 \pm 0.100$ | $0.550 \pm 0.052$ | $\sim 10^5$ | $0.208 \pm 0.018$ |
| RAG (GPT-oss-20b,$k$=5) | $0.597 \pm 0.019$ | $0.564 \pm 0.016$ | $0.850 \pm 0.025$ | $0.678 \pm 0.004$ | $\sim 10^4$ | $0.039 \pm 0.018$ |
| RAG (Llama3-70b,$k$=10) | $0.463 \pm 0.019$ | $0.474 \pm 0.011$ | $0.673 \pm 0.052$ | $0.556 \pm 0.012$ | $\sim 10^5$ | $0.154 \pm 0.114$ |
| RAG (GPT-oss-20b,$k$=10) | $0.598 \pm 0.038$ | $0.563 \pm 0.023$ | $\mathbf{0.880} \pm 0.066$ | $0.687 \pm 0.035$ | $\sim 10^4$ | $0.039 \pm 0.004$ |
| GraphLogDebugger (Ours) | $\mathbf{0.685} \pm 0.065$ | $\mathbf{0.637} \pm 0.082$ | $0.870 \pm 0.099$ | $\mathbf{0.734} \pm 0.024$ | 40.39 | $592.073 \pm 16.513$ |

| | Dataset: **HDFS** Task: **Object Anomaly** | | | | | |
| Method | Detection Effectiveness | | | | Efficiency | |
| | Acc. | Prec. | Recall | F1 | GFLOPs | Throughput (it/s) |
|---|---|---|---|---|---|---|
| MLP | $0.799 \pm 0.053$ | $0.801 \pm 0.052$ | $0.989 \pm 0.000$ | $0.885 \pm 0.032$ | 1.4 | $479.0 \pm 194.0$ |
| RAG (Llama3-70b,$k$=5) | $0.165 \pm 0.022$ | $0 \pm 0*$ | $0 \pm 0*$ | $0 \pm 0*$ | $\sim 10^5$ | $0.162 \pm 0.085$ |
| RAG (GPT-oss-20b,$k$=5) | $0.138 \pm 0.029$ | $0 \pm 0*$ | $0 \pm 0*$ | $0 \pm 0*$ | $\sim 10^4$ | $0.183 \pm 0.010$ |
| RAG (Llama3-70b,$k$=10) | $0.173 \pm 0.040$ | $0 \pm 0*$ | $0 \pm 0*$ | $0 \pm 0*$ | $\sim 10^5$ | $0.194 \pm 0.004$ |
| RAG (GPT-oss-20b,$k$=10) | $0.138 \pm 0.029$ | $0 \pm 0*$ | $0 \pm 0*$ | $0 \pm 0*$ | $\sim 10^4$ | $0.192 \pm 0.027$ |
| GraphLogDebugger (Ours) | $\mathbf{0.989} \pm 0.023$ | $\mathbf{1.000} \pm 0.000$ | $\mathbf{0.987} \pm 0.029$ | $\mathbf{0.993} \pm 0.015$ | 5.57 | $529.999 \pm 201.308$ |

| | Dataset: **Analyst** Task: **Event Anomaly** | | | | | |
| Method | Detection Effectiveness | | | | Efficiency | |
| | Acc. | Prec. | Recall | F1 | GFLOPs | Throughput (it/s) |
|---|---|---|---|---|---|---|
| MLP | $0.948 \pm 0.019$ | $0.922 \pm 0.064$ | $0.980 \pm 0.043$ | $0.950 \pm 0.016$ | 5.58 | $1996.0 \pm 454.0$ |
| RAG (Llama3-70b,$k$=5) | $0.408 \pm 0.038$ | $0.426 \pm 0.026$ | $0.527 \pm 0.029$ | $0.471 \pm 0.019$ | $\sim 10^5$ | $0.204 \pm 0.009$ |
| RAG (GPT-oss-20b,$k$=5) | $0.770 \pm 0.106$ | $0.771 \pm 0.157$ | $0.773 \pm 0.014$ | $0.772 \pm 0.085$ | $\sim 10^4$ | $0.145 \pm 0.018$ |
| RAG (Llama3-70b,$k$=10) | $0.377 \pm 0.014$ | $0.400 \pm 0.004$ | $0.493 \pm 0.038$ | $0.442 \pm 0.014$ | $\sim 10^5$ | $0.204 \pm 0.015$ |
| RAG (GPT-oss-20b,$k$=10) | $0.803 \pm 0.090$ | $0.798 \pm 0.099$ | $0.813 \pm 0.087$ | $0.805 \pm 0.087$ | $\sim 10^4$ | $0.149 \pm 0.014$ |
| GraphLogDebugger (Ours) | $\mathbf{0.957} \pm 0.040$ | $\mathbf{0.921} \pm 0.069$ | $\mathbf{1.000} \pm 0.000$ | $\mathbf{0.959} \pm 0.037$ | 8.97 | $1037.6 \pm 286.2$ |

| | Dataset: **Landslide** Task: **Object Anomaly** | | | | | |
| Method | Detection Effectiveness | | | | Efficiency | |
| | Acc. | Prec. | Recall | F1 | GFLOPs | Throughput (it/s) |
|---|---|---|---|---|---|---|
| MLP | $0.831 \pm 0.079$ | $0.842 \pm 0.180$ | $0.841 \pm 0.149$ | $0.838 \pm 0.056$ | 5.58 | $5391.0 \pm 328.0$ |
| RAG (Llama3-70b,$k$=5) | $0.543 \pm 0.074$ | $\mathbf{0.944} \pm 0.056$ | $0.095 \pm 0.156$ | $0.168 \pm 0.267$ | $\sim 10^5$ | $0.355 \pm 0.044$ |
| RAG (GPT-oss-20b,$k$=5) | $0.611 \pm 0.068$ | $0.701 \pm 0.050$ | $0.389 \pm 0.246$ | $0.495 \pm 0.195$ | $\sim 10^4$ | $0.155 \pm 0.136$ |
| RAG (Llama3-70b,$k$=10) | $0.551 \pm 0.034$ | $0.904 \pm 0.204$ | $0.119 \pm 0.102$ | $0.208 \pm 0.160$ | $\sim 10^5$ | $0.321 \pm 0.193$ |
| RAG (GPT-oss-20b,$k$=10) | $0.623 \pm 0.074$ | $0.723 \pm 0.081$ | $0.397 \pm 0.149$ | $0.511 \pm 0.144$ | $\sim 10^4$ | $0.166 \pm 0.008$ |
| GraphLogDebugger (Ours) | $\mathbf{0.840} \pm 0.080$ | $0.798 \pm 0.117$ | $\mathbf{0.929} \pm 0.059$ | $\mathbf{0.858} \pm 0.062$ | 19.70 | $1334.13 \pm 108.40$ |

**Baselines.** Our framework naturally generalizes to tabular log in different domains. Hence, we mainly compare it to baselines which are generally capable of dynamically processing different types of tabular log that contains text-rich and categorical attributes. **MLP** exploits a pretrained text embedding model to embed events and a learnable embedding for objects. A 3-layer MLP is then applied to map the embeddings to anomaly scores. We also compare a series of baselines based on retrieval augmented generation (**RAG**) (Lewis et al., 2020), which is the mainstream method to process general tabular log in the realistic scenario (Akhtar et al., 2025). We deploy RAG based on two advanced open-sourced LLMs, Llama-3-70b (Dubey et al., 2024) and GPT-oss-20b (Agarwal et al., 2025) with the 5 and 10 retrieval entries. The retrieval database is built on the whole training

split and the seen log entries during the online evaluation. We construct specified prompts for different datasets and omitted the description to Appendix A. Both MLP and all RAG baselines use all-MiniLM-L6-v2 [3] (Reimers & Gurevych, 2019) as the text embedding model. We do not compare to baselines on log anomaly detection because all these methods are either template-based or domain-specific, which cannot be applied to datasets other than computer system log.

**Task.** Following the setting of unsupervised anomaly detection (Liu et al., 2021; Schmidl et al., 2022), our basic task is to output an anomaly score for each log entry $x_n$ at timestamp $t_n$, where higher scores denote more outlyingness (Han et al., 2022). In our task, we use 1 to denote anomalies and 0 to denote normal examples in the ground-truth. We use the first 90% split of the dataset for training, where both the log entries and their anomaly labels are available to access for methods. Methods train the model on this training split or use it for the retrieval database. For the rest 10%, we use it as the test split in our online evaluation, where methods can make use of the seen log entries but their anomaly labels are not accessible. We study two types of anomalies in our experiments: object anomalies and event anomalies. Following the definition in Section 3, we inject object anomalies by swapping an object in the log entry with another existing object. To ensure that historical data contains useful information, we only perturb existing objects in the history. Event anomalies are generated similarly by swapping events. The anomaly rate is set to be 0.05.

**Evaluation.** We calculate the metrics for information retrieval: accuracy, precision, recall, and f1 score for the dataset. We also evaluate the efficiency of different methods by GFlops and throughputs. During evaluation, we notice that LLM-based baselines tend to be very slow in processing speed. Hence, we include all anomaly log entries and 50 random normal entries in a subset and run RAG only on this subset. For other baselines and our methods, we obtain the prediction result for the full test split but only compute the metric on the above subset for fair comparison. We run experiments three times and post the average value of metrics with error bars.

**GraphLogDebugger.** The GNN architecture in GraphLogDebugger is a 3-layer GAT backbone with a two-branch node embedder and an MLP prediction head. The node embedder uses 512-d embeddings for objects and the text embedding of all-MiniLM-L6-v2 (Reimers & Gurevych, 2019) for events, with an MLP to map them into the same space. The embedding size of GAT and the prediction head is also 512. We train the GNN for 10 epochs under the learning rate 0.0001 on Adam and the negative ratio 10 on the training split. Following You et al. (2022), we set the batch size as 1 and use a window length of 100 to accelerate the processing.

## 5.2 MAIN RESULTS

Table 2 shows that **GraphLogDebugger** consistently outperforms both MLP and RAG-based baselines across all tabular log datasets on five tasks, in terms of detection performance and efficiency.

**Effectiveness:** GraphLogDebugger achieves the highest F1 scores across all tasks, outperforming RAG baselines—especially in structurally complex domains like HDFS, where RAG methods fail to detect meaningful anomalies (F1 = 0.0). Specifically, RAG baselines are completely fooled by the anomaly pattern that their predicted labels depend on whether there is an existing record with the same format in the retrieved examples, which does not contribute to a reasonable prediction. Two tasks on the Arxiv dataset are the most difficult, where GraphLogDebugger still beats baselines with a higher precision in not abusing anomaly prediction. Even in semantically rich settings such as Analyst and LandSlide, where RAG baselines are expected to excel, our model surpasses them.

**Efficiency:** RAG approaches exhibit extremely low throughput (typically below 0.3 iterations per second) due to the computational overhead of large language models. In contrast, GraphLogDebugger achieves throughput of at least 500 per second, with significantly lower GFLOPs, enabling real-time anomaly detection in high-throughput environments.

## 5.3 CASE STUDY: WHERE DOES RAG FAIL?

It is natural that GraphLogDebugger yields advantages in efficiency compared to the RAG-baseline, for the latter relies on LLMs with billions of parameters. However, the

| Method | Correlation |
|---|---|
| RAG | -0.1087 |
| GraphLogDebugger | 0.1561 |

---

[3] https://huggingface.co/sentence-transformers/all-MiniLM-L6-v2

leading performance of GraphLogDebugger in detection needs further explanation, while RAG enjoys the general comprehension and reasoning ability of modern LLMs. To this end, we study cases from event anomaly detection of the Arxiv dataset. We choose this task because the degree of event nodes can directly reflect the local graph density of the node-of-interest. We calculate the correlation between event node degrees and the accuracy of GraphLogDebugger and that of RAG(GPT-oss-20b,$k$=10). The result in the table shows that **the accuracy of GraphLogDebugger is positively correlated with the node degree, while the accuracy of RAG is negatively correlated with the node degree.** This indicates that GraphLogDebugger outperforms RAG on event nodes with rich connections with objects, where semantics of these objects are necessary to detect the anomaly.

---

**Case 1: Label=negative, RAG=positive, GraphLogDebugger=negative**
**Title**: "SymbioSim: Human-in-the-loop Simulation Platform for Bidirectional Continuing Learning in Human-Robot Interaction"
**Authors**: "Haoran Chen", "Yiming Ren", "Xinran Li", "Ning Ding", "Ziyi Wang", "Yuhan Chen", "Zhiyang Dou", "Yuexin Ma", "Changhe Tu" (9 objects)
**Reason (RAG)**: The author team composition, research domain mismatch, and unclear collaboration patterns raise suspicions about the coherence of the record.
**Case 2: Label=positive, RAG=negative, GraphLogDebugger=positive**
**Title**: "VERA: Explainable Video Anomaly Detection via Verbalized Learning of Vision-Language Models"
**Authors**: "Shubham Gupta", "Zichao Li", "Tianyi Chen", "Cem Subakan", "Siva Reddy", "Perouz Taslakian", "Valentina Zantedeschi" (7 objects)
**Reason (RAG)**: The record seems coherent, with individual authors' expertise areas aligning with the paper's topic, although the team size is slightly larger than expected.
**Case 3: Label=positive, RAG=positive, GraphLogDebugger=negative**
**Title**: "Transformer$^{-1}$: Input-Adaptive Computation for Resource-Constrained Deployment"
**Authors**: "Yitong Yin" (1 objects)
**Reason (RAG)**: The record consists of a single author, which is consistent with similar papers in the same research domain.

---

We further raise three cases above to investigate when and how GraphLogDebugger and RAG fail. In Case 1, RAG predicts the normal example as abnormal because the limited retrieved examples do not provide enough evidence to prove the coherence of the author team. By contrast, GraphLogDebugger validates overall team consistency by checking the research background of every author, which correctly predicts the negative label. Case 2 is complementary to Case 1, where GraphLogDebugger is able to scan the research interest of every author and detect the anomaly accurately. However, when the connected objects are few, such as in Case 3, GraphLogDebugger may not have enough references based on the graph to make a correct judgment. In similar cases, RAG could then outperform GraphLogDebugger to recognize patterns in the number of authors in the same domain.

These cases provide insights on how graphs can benefit retrieval augmented generation. When the key entry has dense connections with other entries, traditional retrieval based on similarity cannot efficiently include enough entries to enhance the generation quality. With the help of modern embedding models, graphs can be introduced to gather information in these multi-entry scenarios.

## 6 CONCLUSION

We propose a general framework to cover online debugging for heterogeneous tabular logs. By modeling online log debugging as anomaly detection of dynamic graphs, our framework integrates different types of log data into a unified modality by text embedding models, where a dynamic GNN debugs the log through link prediction. Our framework shows good performance in four different datasets while maintaining high efficiency compared to the mainstream RAG-based method.

**Limitation.** Our work explores combining dynamic GNNs and text embedding models to process log data under the online setting, which indicates the potential to accelerate the online process of data streams by a graph-based method. Nevertheless, our experiments mainly show this potential in the bug detection setting. We leave the exploration of online bug correction to future work.

## ETHICS STATEMENT

Our work focuses on detecting inconsistencies in general tabular log data, which enhances the progress of automated log data processing in real-world scenarios. While automation of log processing may raise issues concerning hallucination or fraud reporting, our work does not explicitly introduce new risks compared to existing research.

## REPRODUCIBILITY STATEMENT

All implementation details of our method and baselines are given in Section 5 and Appendix A. We will release at the time of publication.

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

## A APPENDIX

### A.1 USE OF LLMs

We use ChatGPT to polish our introduction (Section 1) and generate the notation table (Table 3), both of which have been checked manually. We also use ChatGPT to retrieve related works in the tabular log processing part by searching machine-learning based log processing methods.

### A.2 NOTATION

Table 3: **Notation**

| Symbol | Type | Meaning |
|---|---|---|
| **Tabular-log basics (Sec. 3)** | | |
| $X = \{x_0, \ldots, x_{N-1}\}$ | sequence | Time-ordered tabular log (entries). |
| $x_n$ | entry | The $n$-th log entry. |
| $t_0 < \cdots < t_{N-1}$ | timestamps | Arrival times of entries. |
| $x_n^m$ | attribute value | The $m$-th attribute in entry $x_n$. |
| $M$ | integer | Number of attributes per entry. |
| $\{o_n^0, \ldots, o_n^{P-1}\}$ | set | Object attributes extracted from $x_n$. |
| $P$ | integer | Number of object attributes in $x_n$ ($P < M$). |
| $s_n$ | text / node | Event attribute (one per entry; possibly text). |
| $\{f_n^0, \ldots, f_n^{Q-1}\}$ | set | Feature attributes extracted from $x_n$. |

*Continued on next page*

| Symbol | Type | Meaning |
|---|---|---|
| $Q$ | integer | Number of feature attributes in $x_n$ ($Q < M$). |
| $y_n$ | label | Event anomaly label for $x_n$ (0 normal, 1 abnormal). |
| $y_n^p$ | label | Object anomaly label for object $o_n^p$ (0/1). |
| $y_n^q$ | label | Feature anomaly label for feature $f_n^q$ (0/1). |
| **Graphs and dynamics (Sec. 4.1–4.2)** | | |
| $\mathcal{G}$ | dynamic graph | Evolving heterogeneous graph over time. |
| $G_n$ | snapshot | Graph snapshot at time $t_n$ (before merging $g_n$). |
| $g_n$ | subgraph | Subgraph constructed from new entry $x_n$. |
| $G_{n+1} \setminus G_n$ | graph diff | Increment between consecutive snapshots; here equal to $g_n$. |
| $\mathcal{V}, \mathcal{E}$ | sets | Node and edge sets of the current graph. |
| $v, e$ | node, edge | A node or an edge (generic). |
| $\mathcal{V}_n^o$ | node set | Object nodes appearing in $x_n$. |
| $\mathcal{V}_n^e$ | node set | New event nodes introduced by $x_n$ (events are unique). |
| $\mathcal{E}_k^+$ | edge set | Positive (observed) object–event links in $g_k$. |
| $\mathcal{E}_k^-$ | edge set | Negative samples (non-existent object–event pairs). |
| $\hat{\mathcal{E}}_n^+$ | edge set | Accepted/predicted-positive links at $t_n$. |
| $\hat{\mathcal{V}}_n^e$ | node set | Accepted new events incident to $\hat{\mathcal{E}}_n^+$. |
| $\mathcal{R}_{t_n}$ | set | Per-time link predictions/results at $t_n$. |
| $\{G_n\}_{n=0}^{N-1}$ | sequence | The sequence of snapshots defining $\mathcal{G}$. |
| $G_{t_K}, G_{t_N}$ | snapshots | Snapshot after train time $t_K$, and final snapshot at $t_N$. |
| **Modeling (GNN and scoring; Sec. 4.3)** | | |
| $\theta$ | parameters | Trainable parameters of the GNN. |
| $f_\theta(\cdot)$ | mapping | GNN that computes object-node embeddings on $\mathcal{G}$. |
| $\mathcal{F}$ | encoder | Text (and time-aware) embedding model for events. |
| $h_o, h_e$ | vectors | Object and event embeddings, respectively. |
| $\text{reduce}(\cdot, \cdot)$ | operator | Embedding combiner (e.g., concat/diff/dot). |
| $\text{MLP}(\cdot)$ | mapping | Multi-layer perceptron used for scoring. |
| $\sigma(\cdot)$ | function | Sigmoid activation. |
| $s_{o,e}$ | score | Link-normality score for pair $(o, e)$. |
| $\hat{\ell}_{o,e}$ | label | Predicted link label: $\mathbb{1}[s_{o,e} \geq \tau]$. |
| $\mathcal{L}_k$ | loss | Balanced BCE loss at training step $k$. |
| $\eta$ | scalar | Learning rate. |
| $\tau$ | threshold | Operating threshold for prediction. |
| $\rho$ | ratio | Negative sampling ratio. |
| **Data splits and indices** | | |
| $X_{\text{train}}, X_{\text{test}}$ | sequences | Training and test splits (chronological). |
| $K$ | integer | Index/time that separates train and test. |
| $N$ | integer | Total number of entries/snapshots. |
| $k, n$ | indices | Training step $k$, evaluation time $n$. |
| $t_k, t_n$ | timestamps | Times associated with steps/entries. |

### A.3   MODEL DESIGN SPACE

We compare two variants in our experiments: (i) Plain (ungated) GAT. We first concatenate the entity-type and entity-ID embeddings and pass them through a feed-forward projection to obtain the initial representation $e_0$. We then run multi-layer, multi-head GATConv on an entity–entity graph induced by shared content to propagate messages and obtain $e_{\text{GAT}}$, which we use as the final entity representation. (ii) Gated fusion. Starting from the same $e_0$ and $e_{\text{GAT}}$, we introduce a global learnable scalar gate $\alpha$ and adaptively combine them via a sigmoid: $e = (1 - \sigma(\alpha)), e_0 + \sigma(\alpha), e_{\text{GAT}}$. This biases toward $e_0$ when the given signal is weak (or absent) and toward $e_{\text{GAT}}$ when the signal is strong. Both variants share the same link-prediction head: we take the entity representation and the content representation (text and time embeddings concatenated and then projected), compute their element-wise difference, and feed it to an MLP to output the link probability

## A.4 Additional Visualization

Figure 3 visualizes the distribution of anomaly likelihood scores of our five evaluation tasks. The score distribution corroborates the main result in Table 2, that Analyst, Arxiv (Node), and HDFS are three tasks relatively easy, with the score distribution of anomalies and normal examples separate clearly. By contrast, the score of anomalies and normal examples mix up in Arxiv (Edge) and Landslide, indicating that these datasets are more difficult.

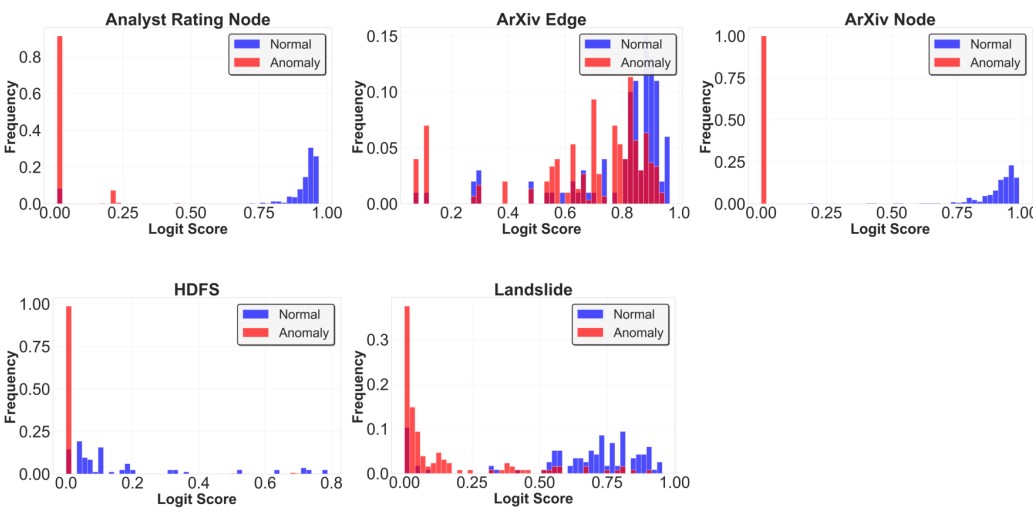

Figure 3: **Anomaly score distribution of five tasks by GraphLogDebugger.** Score distributions of anomalies and normal examples separate for simpler tasks and mix up for more difficult tasks.

## A.5 Prompts in RAG

We list the prompt we used in our RAG baseline as follows:

```
1  """Build context for ArXiv dataset (authors and paper titles)."""
2  context = """You are an expert at analyzing author-paper relationships in
        academic research.
3
4  DATASET CONTEXT: This is a dataset of academic papers with their authors
        and titles.
5  - Entities (authors): Research authors who wrote the papers
6  - Content (titles): The titles of the academic papers
7  - Edge: A connection between an author and a paper title (indicating the
        author contributed to that paper)
8
9  TASK: Determine if the specific author-paper connection (edge) should
        exist based on historical patterns.
10
11  EDGE ANALYSIS TARGET:
12  """
13
14  context += f"Author: {entity_name}\n"
15  context += f"Paper Title: {content_name}\n\n"
16
17  if similar_contents:
18      context += "SIMILAR PAPERS AND THEIR AUTHORS (for reference):\n"
19      context += "Use these examples to understand what types of authors
            typically work on similar papers.\n\n"
20
```

```python
21      for i, content_record in enumerate(similar_contents[:10]):
22          content = content_record.get('content', '')
23          entities = content_record.get('related_entities', [])
24          similarity = content_record.get('similarity', 0.0)
25          num_records = content_record.get('num_records', 0)
26
27          context += f"{i+1}. Paper Title: {content} (Similarity: {
                similarity:.3f}, {num_records} records)\n"
28          context += f"   Authors who worked on this paper: {', '.join(
                entities) if entities else 'None'}\n\n"
29  else:
30      context += "No similar papers found in historical data.\n\n"
31
32  context += """ANALYSIS QUESTION:
33  Based on the similar papers and their author patterns, should the
        specified author-paper connection exist?
34
35  EVALUATION CRITERIA:
36  1. Research Domain Match: Does the author's expertise align with the
        paper's topic?
37  2. Historical Patterns: Do authors with similar expertise appear in
        similar papers?
38  3. Authorship Likelihood: Is it reasonable that this author would
        contribute to this type of research?
39  4. Anomaly Detection: Does this connection seem unusual or out of place
        compared to patterns in similar papers?
40
41  DECISION GUIDELINES:
42  - edge_exists = True: The author-paper connection makes sense based on
        research area and historical patterns
43  - edge_exists = False: The author seems misplaced or unlikely to work on
        this type of paper (anomalous edge)
44  - Consider the research fields, methodologies, and typical author
        patterns shown in similar papers
45  - An edge is anomalous if the author appears completely unrelated to the
        research domain of the paper
46
47  CONFIDENCE SCORING:
48  - High confidence (0.8-1.0): Clear patterns in similar papers strongly
        support/reject the connection
49  - Medium confidence (0.5-0.7): Some evidence but less certain
50  - Low confidence (0.0-0.4): Limited historical data or unclear patterns
51  """
```

Listing 1: Prompt: Arxiv

```python
1   """Build context for HDFS dataset (BlockId focus for detection)."""
2   context = """You are an expert at analyzing BlockId-log relationships in
        HDFS distributed file system logs.
3
4   DATASET CONTEXT: This is a dataset of HDFS system logs with their Block
        IDs and log contents.
5   - Primary Focus: Block IDs (e.g., blk_8215417782549978040,
        blk_161475555609545016) - unique identifiers for HDFS data blocks
6   - Content (logs): The actual log messages and operations in the HDFS
        system that involve specific blocks
7   - Edge: A connection between a Block ID and a log message (indicating the
        block is involved in that log operation)
8
9   SPECIAL NOTE: For HDFS anomaly detection, we focus specifically on Block
        ID connections to log messages.
10  Block IDs should appear BOTH in the BlockId column AND within the log
        content itself.
11
```

```
12 TASK: Determine if the specific Block ID-log connection (edge) should
       exist based on historical patterns.
13
14 EDGE ANALYSIS TARGET:
15 """
16
17 context += f"Block ID: {entity_name}\n"
18 context += f"Log Content: {content_name}\n"
19 context += f"Content Analysis: Does '{entity_name}' appear in the log
       content? {'YES' if entity_name in content_name else 'NO'}\n\n"
20
21 if similar_contents:
22     context += "SIMILAR LOG MESSAGES AND THEIR BLOCK IDs (for reference)
           :\n"
23     context += "Use these examples to understand what types of Block IDs
           typically appear in similar log messages.\n\n"
24
25     for i, content_record in enumerate(similar_contents[:10]):
26         content = content_record.get('content', '')
27         entities = content_record.get('related_entities', [])
28         similarity = content_record.get('similarity', 0.0)
29         num_records = content_record.get('num_records', 0)
30
31         block_ids = [e for e in entities if e.startswith('blk_')]
32         other_entities = [e for e in entities if not e.startswith('blk_')
               ]
33
34         context += f"{i+1}. Log Content: {content} (Similarity: {
               similarity:.3f}, {num_records} records)\n"
35         context += f"   Block IDs in this log: {', '.join(block_ids) if
               block_ids else 'None'}\n"
36         if other_entities:
37             context += f"   Other entities: {', '.join(other_entities
                   [:3])}{'...' if len(other_entities) > 3 else ''}\n"
38         context += "\n"
39 else:
40     context += "No similar log messages found in historical data.\n\n"
41
42 context += """ANALYSIS QUESTION:
43 Based on the similar log messages and their Block ID patterns, should the
       specified Block ID-log connection exist?
44
45 EVALUATION CRITERIA:
46 1. Block ID Presence: Does the Block ID appear within the log content
       itself? (This is crucial for HDFS)
47 2. Log Operation Match: Does the Block ID relate to the HDFS operation
       described in the log?
48 3. Historical Patterns: Do similar Block IDs appear in similar log
       messages?
49 4. HDFS Block Behavior: Is it reasonable that this Block ID would be
       involved in this type of operation?
50 5. Content Consistency: Block ID should be consistent between the BlockId
       column and the log content
51
52 DECISION GUIDELINES:
53 - edge_exists = True: The Block ID-log connection makes sense based on
       HDFS block operations and historical patterns
54 - edge_exists = False: The Block ID seems unrelated to this log message (
       anomalous edge)
55 - CRITICAL: If the Block ID does NOT appear in the log content, this is
       likely anomalous
56 - Consider HDFS block operations like allocation, storage, replication
       shown in similar messages
57 - An edge is anomalous if the Block ID appears completely unrelated to
       the log operation
```

```
58
59 CONFIDENCE SCORING:
60 - High confidence (0.8-1.0): Clear Block ID patterns and content
       consistency strongly support/reject the connection
61 - Medium confidence (0.5-0.7): Some evidence but less certain about Block
        ID relevance
62 - Low confidence (0.0-0.4): Limited historical data or unclear Block ID
       patterns
63
64 IMPORTANT: Focus specifically on Block ID relationships - Components and
       Event IDs are secondary for this analysis.
65 """
```

Listing 2: Prompt: HDFS

```
1 """Build generic context for unknown datasets."""
2 context = f"""You are an expert at analyzing entity-content relationships
      .
3
4 EDGE ANALYSIS TARGET:
5 Entity: {entity_name}
6 Content: {content_name}
7
8 TASK: Determine if this entity-content connection should exist based on
      historical patterns.
9 """
10
11 if similar_contents:
12     context += "\nSIMILAR EXAMPLES:\n"
13     for i, content_record in enumerate(similar_contents[:5]):
14         content = content_record.get('content', '')
15         entities = content_record.get('related_entities', [])
16         context += f"{i+1}. Content: {content}\n   Related entities: {',
              '.join(entities)}\n\n"
17
18 context += """
19 DECISION: Should this entity-content connection exist?
20 - edge_exists = True: The connection makes sense based on patterns
21 - edge_exists = False: The connection seems anomalous
22 """
```

Listing 3: Prompt: Analyst and Landslide

