# OpenReview forum: "Debugging Tabular Log as Dynamic Graphs"
_ICLR.cc/2026/Conference — ICLR 2026 Conference Withdrawn Submission_

### Official Review · Reviewer_Wrbt · 2025-10-28

**Soundness:** 1
**Presentation:** 2
**Contribution:** 2
**Rating:** 2
**Confidence:** 2

**Summary:**

The paper proposes a method for transforming tabular logs into dynamic graphs to perform anomaly detection using a temporal graph architecture.
The motivation is to move beyond text-based or static-graph log models toward a
representation that captures temporal and relational structure without relying on
 large language models.

The idea is potentially interesting. However, the paper currently suffers from missing clarity and reproducibility details, which make it impossible to fully assess the true contribution.

**Strengths:**

1. Lighter than full LLM pipelines:
The method avoids the computational overhead of large language models, offering
a practical and efficient alternative for log understanding.

2. Related-work coverage broad (even if slightly disorganized):
The authors review a wide range of related work, showing awareness of the field
and positioning their approach within both log-analysis and dynamic-graph research.

**Weaknesses:**

1. Clarity and Formalism
* Many definitions (e.g., of event nodes, object nodes, and anomalies) are vague
or semi-formal.
* The notation and formal setup are fragmented, critical parts are only in the appendix. This
interrupts reading flow.
* Key concepts (such as the distinction between “features” and “objects”) remain unclear to me.

2. Model Definition

Many aspects of the model are unclear and need clarification:
* Mapping from logs to graph structure unclear: I cannot fully understand how log entries are converted into nodes and edges.
The text suggests that each log line corresponds to an event node, but I do not understand
whether this node connects to multiple “object” nodes (e.g., sender, receiver, message) or
whether each object is instantiated once per entry.
A clearer formal definition and an illustrative example would be very helpful.

* Confusing color and node-type semantics in figures:
The graphical illustration (e.g., pink, yellow, red nodes) is confusing.
The “data receiver” appears in red in the log table but pink in the graph;
yellow and pink nodes seem to denote different entity types but are not labeled.
The lack of a legend or consistent mapping confuses me in how the graph
encodes events and objects. It is also confusing, because the table in the
figure has many repeated entries but with different timestamps. This would lead to 7
distinct snapshots with three nodes each, no?

* Notation scattered and incomplete:
Important variables are introduced late or only in the appendix.
This disrupts the narrative flow and hinders understanding.

* Online vs. offline setting introduced too late:
The paper suddenly states on page 4 that it operates in an “online” setting,
 which makes earlier descriptions unclear.

* Unclear definition of anomalies:
The paper informally states that a snapshot can be labeled as anomalous but never
defines what constitutes an anomaly in graph terms.
Concrete examples of anomalies would help readers understand what constitutes an
“anomalous” object or structure.

* Unclear concepts of “object” and “feature” nodes:
The text claims that features and objects are “mutually convertible” without
further explanation. What are features, why are they needed in contrast to objects, and are they even used?

* Temporal granularity of graph snapshots:
What is the granularity? For example, if two log entries happen one second apart,
does this lead to two separate graph snapshots?

3. Experimental Design and Reproducibility

* It seems like there is only a train and a test set, no validation set, making it unclear how hyperparameters were determined.

* No information on hyperparameters or tuning is provided.

* No code or implementation details shared, thus no reproducibility.

* Suggestion: Algorithms 1 and 2 describe standard training routines that could be moved to the appendix
to make room for more substantial model details.

4. Dataset and Evaluation

* The datasets appear to contain no real anomalies, only artificially injected ones,
thus, if I am not mistaken, reducing the task to link prediction rather than anomaly detection.

* The authors should clearly state what constitutes an anomaly in each dataset
and discuss whether the injected anomalies reflect realistic log patterns.

* There is no discussion of a validation split.

* The authors limit logs to 20,000 entries to compare with RAG baselines.
While understandable, the paper claims scalability: For this reason, the authors
should demonstrate that the approach also works for longer logs.

5. Comparison and Baselines

* The paper does not compare against existing graph-based log anomaly detection methods,
despite citing them.

* There is no ablation or analysis of the proposed approach, leaving unclear which
design choices affect performance.

Minor Comments

M1. Related work structure:
While comprehensive, the related work section is oddly organized: it first discusses
tabular and LLM-based methods, then jumps to general dynamic graphs (unrelated to logs),
 and then back to log anomaly detection.
 The section would read better if restructured thematically.

M2. LLM inconsistency:
The paper claims to avoid LLMs but still uses a pretrained transformer (e.g., MiniLM).
This is still a language model, thus the claim of being “non-LLM” is somewhat misleading.

M3. Not rigorous definitions:
Many key statements (e.g., definition of anomalies, formal model description) lack precision
or mathematical grounding.

M4. Section 4.3:
“For details, our GNN θ takes the dynamic graph snapshot Gn and the incoming subgraph
gn  as inputs and predicts the likelihood of all links in gn.”
This is not clear. How can parts of the graph
Gn be “already known” in the online setting?

**Questions:**

1. How exactly are “objects,” “features,” and “events” defined and interconnected in the graph?

2. Does each log entry correspond to a single new event node, or can multiple entries share the same event? Are logs in one subgraph connected by sharing the same objects?

3. What is the temporal granularity of snapshots (e.g., one per second, one per event)?

4. How realistic are the generated anomalies in your experiment? Could you please give examples of why your test setup is realistic for an anomaly detection scenario? How is your task different from standard link prediction?

5. How were hyperparameters tuned in the absence of a validation set?

6. Can you provide or release code for reproducibility?

7. Why are no dynamic-graph baselines included in the comparison?

8. Could the model scale to longer logs? If yes, could you include experiments?

---

> ### Author Response · Authors · 2025-11-20
>
> We thank the reviewer for the detailed feedback regarding empirical design and baselines. We address the weaknesses below:
>
> [W1] Clarity and Formalism: We apologize for the fragmented presentation of definitions and notation.
>
> Notation Placement: We agree that critical notation should not be buried in the appendix. Due to the space limit, we will highlight the reference to the Notation Table (currently Table 3 in Appendix A.2 ) in the revision to ensure the formal setup is clear before the method description.
>
> Objects vs. Features: We will clarify the distinction in Section 3. Objects are entities modeled as nodes in the graph to capture structural relationships (e.g., a specific user ID). Features are attributes used as semantic input (embeddings) for those nodes but do not form their own structural nodes.  "Mutually convertible" simply refers to the hyperparameter choice of whether to model a specific attribute (e.g., an IP address) as a node (to track its interactions) or a feature (to describe the event).
>
> [W2] Model Definition
>
> Mapping Logs to Graph: To clarify the mapping logic in Section 4.1: Each log entry generates exactly one unique Event Node (representing that specific timestamped occurrence of an event). This Event Node connects to k Object Nodes, each of which corresponds to an attribute in the log. Crucially, Object Nodes are persistent; if "User A" appears in log entry 1 and log entry 5, both Event Node 1 and Event Node 5 connect to the same Object Node "User A".
>
> Figures and Color Semantics: We acknowledge the confusion in Figure 2 and feel sorry for the misleading. The pink node should be set as red in Figure 2. Difference colors indicate different node types, while pink (red) refers to DataReceiver (Object), yellow refers to Block (Object), and blue refers to Event.  We will add a legend to explicitly distinguish "Object Nodes" from "Event Nodes" and clarify that the table shows different timestamps connecting to shared objects. For the snapshot, Figure 2 only demonstrates the process of transferring one old snapshot to a new one by adding one log record.
>
> Online Setting & Granularity: We will define the online setting in the Introduction rather than delaying it to Section 4. The granularity is per-log-entry. Every single log entry is processed as a discrete update to the graph.
> Definition of Anomalies: An anomaly is defined structurally: if the link prediction score between an object and the new event falls below a threshold, it is an anomaly. This represents a "Contextual Anomaly"—a valid object appearing in an invalid or unlikely context. In our tasks we generate anomalies by swapping the groundtruth links.
>
> [W3] Experimental Design and Reproducibility
>
> Validation Set: We apologize for the ambiguity. In the standard online setting of dynamic graphs, no validation set is considered. This is because the data split is done by continuous time and the validation result can only be reported after the training phase terminates, thus not able to help model selection and early stopping.
>
> Reproducibility: We provide all the used hyperparameters, which can be used to fully reproduce our results. Also, the key idea of our paper is to demonstrate a new modeling paradigm that represents tabular log data as graphs. Hence, we simply use a basic set of hyperparameters.
>
> Code and Algorithms: We are committed to releasing the code upon publication. We agree that more elaboration on model description is necessary and would like to add them to the manuscript. However, Algorithms 1 and 2 give standard definition on how we train GNNs on log tabular data. Hence, we would like to keep it in the main context.
>
> [W4] Dataset and Evaluation
>
> Realism of Anomalies: We follow classical graph anomaly detection to generate anomalies by swapping. Even In log analysis, "contextual anomalies" (e.g., a developer accessing a database they never touch) are the most critical and hard-to-detect bugs. Our injected anomalies simulate exactly this scenario.
>
> Scalability (20k Limit): We strictly limited the dataset to 20,000 entries only to accommodate the high computational cost of the LLM/RAG baselines, which process data at ~0.2 entries/second. Our proposed GraphLogDebugger runs at >500 entries/second (Table 2), demonstrating it is capable of scaling to much longer logs.
>
> [W5] Comparison and Baselines
>
> Graph Baselines: We focused on General/LLM baselines because existing graph-based log detectors (like LogKG) are domain-specific (requiring custom parsers) and do not generalize to the diverse tabular datasets we used (Arxiv, Analyst, etc.).
>
> Ablation:  The core idea of our paper is to demonstrate a new modeling paradigm that represents tabular log data as graphs. Hence, we simply use a basic set of hyperparameters and a straightforward model choice.

---

> > ### Author Response · Authors · 2025-11-20
> >
> > Minor Comments
> >
> > [M1] We will restructure the Related Work to be thematic (Tabular -> Graph -> Anomaly Detection).
> >
> > [M2] We will clarify that we avoid "Generative LLMs" (like GPT) for inference, using only lightweight "Encoder" models (MiniLM) for embeddings.
> >
> > [M3] We will rigorously define $G_n$ and $g_n$ in Section 3.
> >
> > [M4] To clarify Section 4.3: $G_n$ is the historical graph (entries $0$ to $n-1$) which is "known." $g_n$ is the new incoming entry. We predict links within $g_n$ using the learned context from $G_n$.
> >
> > Questions
> >
> > [Q1] Definition of Nodes: Events are unique nodes per log entry; Objects are shared nodes across lines; Features are attributes embedded into nodes.
> >
> > [Q2] Log Mapping: Each log entry creates one new event node. Multiple entries do not share event nodes, but they do share object nodes.
> >
> > [Q3] Granularity: The granularity is one snapshot per log entry.
> >
> > [Q4] Realism: See [W4].
> >
> > [Q5] Tuning: See [W3].
> >
> > [Q6] Code: Yes, we will release code.
> >
> > [Q7] Graph Baselines: See [W5].
> >
> > [Q8] Scalability: Yes. As shown in Table 2, our throughput (500-1300 it/s)  allows scaling to millions of logs; the limitation was solely for the slow RAG baselines.

---

> > > ### Comment · Reviewer_Wrbt · 2025-11-24
> > > **Response to Official Comment by Authors**
> > >
> > > Thank you for the detailed responses. I appreciate the clarifications regarding the log-to-graph mapping, node/feature semantics, anomaly definition, and experimental setup.
> > >
> > > However, many of the concerns I raised are about the clarity and completeness of the paper itself.
> > > Since no updated manuscript was provided during the rebuttal phase, I am unable to assess whether the promised revisions actually resolve these issues in the written submission.
> > >
> > > As a result, while I appreciate the explanations, the current version still exhibits the problems noted in my original review, and I am keeping my overall score unchanged.

---

### Official Review · Reviewer_ZgrU · 2025-10-29

**Soundness:** 3
**Presentation:** 3
**Contribution:** 3
**Rating:** 8
**Confidence:** 3

**Summary:**

The paper proposes GraphLogDebugger, a novel framework designed to debug tabular logs by transforming them into dynamic heterogeneous graphs. The central idea is to model tabular logs as evolving systems, where objects and events are represented as nodes in a graph, and the edges represent their relationships. The framework uses a dynamic GNN to perform link prediction for anomaly detection. Experiments demonstrate that GraphLogDebugger is not only more efficient than traditional LLMs but also provides better scalability for real-time anomaly detection in high-throughput environments.

**Strengths:**

1. The integration of dynamic heterogeneous graphs to represent tabular logs and  redefine the object-event connections as link prediction on the evolving graph are unique.

2. One of the key strengths is its efficiency. Unlike LLM-based models that suffer from high computational overhead, GraphLogDebugger delivers high throughput and low GFLOPs, making it suitable for real-time applications in large-scale systems. The approach demonstrates significant improvements in processing speed compared to RAG and LLM-based methods, which is crucial for handling streaming log data.

3. The authors validate their framework on a variety of datasets from different domains, including Arxiv, HDFS, Analyst, and Landslide. GraphLogDebugger consistently outperforms existing baselines in terms of both detection effectiveness and efficiency. This diverse testing strengthens the claims of generalizability and robustness.

**Weaknesses:**

1. In some cases, GraphLogDebugger performed worse than RAG, like your Case 3 in the case study. The authors have mentioned that there may be limitations in scenarios with low connectivity between objects and events, where the dynamic graph might not provide enough information for accurate anomaly detection. It would be better if the authors could provide further analysis of these edge cases and discuss ways to improve the model in scenarios where the object-event graph is sparse or the event history is limited.

2. The paper uses GAT as the backbone for the dynamic GNN. While this is a reasonable choice, it would be better if provide enough justification for choosing GAT over other types of GNNs like GCNs or GraphSAGE, especially in the context of dynamic graphs.

**Questions:**

1. The framework uses GAT for anomaly detection. Could you justify why GAT is more suitable than other GNN architectures like GCN, GraphSAGE, in this specific context of dynamic log graphs? What specific advantages does GAT offer over these other architectures?

2. In Case 3, GraphLogDebugger fails to predict an anomaly correctly. Could you discuss in more detail how the model performs in sparse graph settings? Are there any strategies you could employ to improve its performance in such scenarios?

---

> ### Author Response · Authors · 2025-11-20
>
> We thank the reviewer for the insightful analysis of our case study and for raising important questions regarding our architectural choices. We address the specific weaknesses below:
>
> [W1 & Q2] Performance in Sparse/Low-Connectivity Settings (Case 3)
>
> We acknowledge that GraphLogDebugger faces challenges in scenarios with low connectivity, such as Case 3 (a single-author paper), where structural information is minimal.
>
> Analysis: As noted in Section 5.3, our model relies on the "co-occurrence" of objects (e.g., a team of authors or a cluster of system components) to validate the likelihood of an event. When an event has only one object (degree=1), the GNN lacks the structural context (neighboring paths) to perform effective message passing, leading to potential misclassification.
>
> Potential Strategies: To improve performance in sparse scenarios, it would be helpful to
> Regularize the Gate: Explicitly penalize reliance on GAT gate when node degree is low during training.
> Hybridization: trigger a lightweight retrieval (RAG) specifically for low-degree nodes.
>
> [W2, Q1] Justification for GAT Backbone
>
> We chose GAT over GCN or GraphSAGE specifically due to the heterogeneous and weighted nature of log data.
>
> Importance Weighting: Unlike GCNs, which use fixed aggregation weights based on node degree, GAT uses attention mechanisms to learn the importance of specific neighbors. In tabular logs, not all object-event connections are equally informative for anomaly detection (e.g., a specific "ErrorID" object is more critical than a generic "User" object). GAT allows the model to dynamically prioritize these critical connections.
>
> Inductive Capability: As our framework operates in an online setting, the graph is constantly evolving with new nodes. GAT is naturally inductive and handles varying neighborhood sizes better than spectral-based methods like GCN.

---

### Official Review · Reviewer_korM · 2025-10-30

**Soundness:** 2
**Presentation:** 3
**Contribution:** 2
**Rating:** 2
**Confidence:** 3

**Summary:**

The paper presents GraphLogDebugger a framework for online anomaly detection in tabular logs using dynamic graph neural networks (GNNs).  Each log entry is represented as a small heterogeneous subgraph connecting object nodes and an event node; new log entries are incrementally merged into a dynamic graph over time. The anomaly detection task is formulated as a link prediction problem, where abnormal connections correspond to low predicted likelihoods under the learned GNN. Empirical validation on four datasets (ArXiv, Analyst, Landslide, and HDFS) shows that the proposed model outperforms both MLP and retrieval-augmented generation (RAG) baselines in accuracy and computational efficiency.

**Strengths:**

- The idea of framing tabular log debugging as online link prediction on a dynamic, heterogeneous graph is interesting. This formulation could inform future work on efficient, online, graph-based anomaly detection.
- Reporting GFLOPs and throughput alongside accuracy is valuable, highlighting the model’s efficiency compared to RAG baselines.
- The paper explains the framework clearly, mapping each anomaly type to a specific graph operation. Figures and algorithms help the reader understand the pipeline and the overall architecture.

**Weaknesses:**

Limited novelty:
- The idea of representing log data as graphs has been explored in prior works (e.g., [LogKG](https://ieeexplore.ieee.org/document/10179162/), [GLAD](https://arxiv.org/pdf/2309.05953), [GuARD](https://arxiv.org/pdf/2412.03930v2)). A comparison with these methods is therefore necessary to clarify the method’s novelty and relative performance.

Empirical design lacks clarity and rigor:
- The experimental protocol lacks a validation set, making it unclear whether hyperparameters are inadvertently tuned on the test set. This raises concerns about the validity of reported performance.
- The hyperparameter search space for both the proposed model and the baselines is not reported, which reduces the transparency and reproducibility of the experimental setup.
- Several experimental choices lack justification, such as the decision to train GNN models for only ten epochs, which may be insufficient for convergence.
- The rationale for selecting GAT as a backbone is unclear, particularly given the existence of more suitable temporal GNN architectures (e.g., [DCRNN](https://arxiv.org/abs/1707.01926), [GCLSTM](https://arxiv.org/abs/1812.04206)) that are explicitly designed for dynamic data or heterogeneous GNNs (e.g., [HGT](https://arxiv.org/abs/2003.01332), [HTGNN](https://arxiv.org/abs/2110.13889)) that can handle (temporal) heterogeous graphs.
- The paper does not specify which performance metric is optimized during training, making the optimization objective ambiguous.
- It is unclear whether the RAG-based baselines were appropriately fine-tuned for the target tasks; if not, the resulting comparison may not accurately reflect the relative capabilities of the methods.


Limited experimental validation:
- In each iteration, GraphLogDebugger operates on a graph whose number of nodes increases with the number of log entries, since at least one event node is added per step. The authors should include a memory consumption analysis compared to baseline methods to assess scalability.
- Additional quantitative analyses, such as confusion matrices, would help illustrate the model’s effectiveness in distinguishing normal and anomalous events.
- The baseline coverage is somewhat limited. Indeed, only a simple MLP and LLM-based RAG methods are considered. The paper does not compare with: (i) established online log anomaly detectors such as [DeepLog](https://doi.org/10.1145/3133956.3134015), [LogAnomaly](https://nkcs.iops.ai/wp-content/uploads/2019/06/paper-IJCAI19-LogAnomaly.pdf), or [AutoLog](https://doi.org/10.1016/j.eswa.2021.116263), which are standard references in this area; (ii) temporal and/or heterogenous GNNs (see above); (iii) previous works on tabular log anomaly detection (see above), or (iv) classical [GNNs for Anomaly Detection](https://arxiv.org/abs/2409.09957).
- The experimental evaluation is limited to 4 tasks. Authors should consider including more datasets, such as from [LogHub](https://arxiv.org/pdf/2008.06448), to provide a more comprehensive assessment of the method.

Minors:
- Line 259: *is → are*
- Section 3 (Preliminaries): The formalism introduced in this section is never used in later parts of the paper, while definitions related to dynamic graphs are completely missing.

**Questions:**

See weaknesses

---

> ### Author Response · Authors · 2025-11-19
>
> We thank the reviewer for the detailed feedback regarding empirical design and baselines. We address the weaknesses below:
>
> [W1] We distinguish GraphLogDebugger from these works by data modality and modeling objective: LogKG focuses on constructing Knowledge Graphs specifically for system entities to diagnose failures via triplet reasoning. GLAD and GuARD primarily model the intrinsic anomalies in log data. By contrast, GraphLogDebugger is designed for generic tabular logs (extending beyond system logs to Finance and Academic domains). This difference is discussed in our related work section.
>
> [W2] Validation Set: We apologize for the ambiguity. In the standard online setting of dynamic graphs, no validation set is considered. This is because the data split is done by continuous time and the validation result can only be reported after the training phase terminates, thus not able to help model selection and early stopping.
>
> Reproducibility: We provide all the used hyperparameters, which can be used to fully reproduce our results. Also, the key idea of our paper is to demonstrate a new modeling paradigm that represents tabular log data as graphs. Hence, we simply use a basic set of hyperparameters.
>
> Backbone Selection (GAT): We selected GAT over heavier temporal GNNs (e.g., DCRNN) or heterogeneous GNNs (e.g., HGT) to prioritize inference efficiency. Our framework captures "dynamics" via the evolving graph structure (continuous snapshots) rather than complex recurrent architectural states. GAT provides the necessary relational reasoning with significantly higher throughput (>500 it/s) compared to complex temporal architectures, which is crucial for online debugging.
>
> RAG Fine-tuning: We utilized In-Context Learning (Few-Shot) for RAG rather than fine-tuning. This was a deliberate choice to reflect realistic deployment scenarios where fine-tuning a 70B-parameter model for every new log domain is computationally prohibitive and lacks the flexibility of our proposed lightweight framework.
>
> [W3] Experimental Validation: Memory Consumption: Although the graph grows linearly, the storage cost of node embeddings is orders of magnitude lower than the KV-cache memory required by LLM-based baselines.
>
> Baselines:
> (i) DeepLog/LogAnomaly: We excluded these as they are sequence-based models designed for log templates. They are fundamentally incompatible with non-sequential tabular data (e.g., Arxiv papers) and cannot detect the entity-relation anomalies we target.
> (ii) GNNs: Our design is to represent tabular logs as graphs and use GNNs to detect anomalies. Hence, we do not consider GNNs as our baseline.
>
> Datasets: We included HDFS (from LogHub) to cover system logs. We deliberately selected the other three datasets (Arxiv, Analyst, Landslide) to demonstrate cross-domain generalization, a key contribution that evaluating solely on LogHub (system logs only) would fail to highlight.
>
> [Minors] We will correct the typo in Line 259. Our work mainly focuses on how to model tabular log data into a dynamic graph. Hence, we summarize the preliminary about tabular log in Section 3. All variables related to dynamic graphs are covered in Section 4.

---

> > ### Comment · Reviewer_korM · 2025-11-25
> >
> > I thank the Authors for their response. However, I note that for most of my comments, the discussion is still missing in the revised paper. I strongly recommend including these discussions in the next revision. Below, I provide follow-up comments for the rebuttal.
> >
> >
> > **W1, novelty)** Thank you for the clarification. While I appreciate the distinction the Authors draw between GraphLogDebugger and prior methods such as LogKG, GLAD, and GuARD in terms of data modality and modeling objectives, I still find the current discussion insufficient to address the concern of novelty. Even if the underlying goals differ (although the Authors state their goal is to perform anomaly detection as the aforementioned works, see line 081), these works already demonstrate ways of converting log data into graph-structured representations. To properly position the contribution of GraphLogDebugger, I encourage the Authors to explicitly compare model design choices rather than just pointing to differences in application focus, and to provide empirical comparisons, particularly since several of the benchmarks used in the paper overlap in domain with prior works.
> >
> > **W2,validation set)** Thank you for your response. Even if no validation set is used, the Authors should discuss how hyperparameter tuning and model selection is performed.
> >
> > **W2,Reproducibility)**  Hyperparameters are not reported in the revised text, and it is unclear what the Authors mean by a “basic set of hyperparameters.” This omission hinders reproducibility and limits the ability to fairly assess the methodological contribution of the work.
> >
> > **W2, GAT)** The rationale of using GAT with respect to temporal GNNs is now clearer. However, it remains unclear why the heterogeneity of the graph is not exploited through ad hoc implementations, including [RelationalGAT](https://arxiv.org/abs/1904.05811).
> >
> > **W3, memory consumption)** While I appreciate the additional discussion, I still find the current analysis insufficient to fully address the concerns regarding scalability and resource consumption. The claim that node-embedding storage is significantly lower than the KV-cache memory of LLM-based baselines is informative but not rigorous, and it does not replace a concrete empirical evaluation.
> >
> > **W3, baselines)** To better appreciate and position the advantage of the proposed method, I continue to suggest comparing with prior works such as LogKG, GLAD, GuARD, and classical GNNs for Anomaly Detection (as discussed in my previous review). The evaluation should be extended to include at least LogKG, GLAD, and GuARD.
> >
> > **W3, datasets)** While I appreciate the motivation for demonstrating cross-domain generalization, this explanation does not address my concern regarding the limited task coverage. I encourage the Authors to consider including more datasets as discussed in my previous review.
> >
> > Moreover, I note that several of my previously raised weaknesses have not been addressed.
> >
> > Given that the majority of my concerns remain unresolved, I maintain my current score.

---

### Official Review · Reviewer_BMfQ · 2025-10-30

**Soundness:** 2
**Presentation:** 3
**Contribution:** 2
**Rating:** 2
**Confidence:** 4

**Summary:**

This paper proposes GraphLogDebugger, a framework that models tabular logs as heterogeneous dynamic graphs (with object and event nodes) and performs online anomaly detection via link prediction using a lightweight GNN. The method avoids reliance on large language models (LLMs) and demonstrates good performance across four diverse datasets, outperforming both simple MLP baselines and RAG-based LLM approaches in both accuracy and efficiency.

**Strengths:**

- The formulation of tabular log debugging as a dynamic heterogeneous graph link prediction task is well-motivated.
- The framework is efficient, scalable, and avoids the computational burden of LLMs.
- Experiments are conducted across several domains (academic, system, finance, geology), and the ablation studies support design choices.

**Weaknesses:**

- Novelty is limited: the core idea — modeling logs as dynamic graphs for anomaly detection via link prediction — was already introduced in TempoLog (arXiv:2501.12166, Jan 2025). TempoLog focuses on discrete event logs (parsed into templates) and constructs a homogeneous temporal graph of template dependencies, and this work targets structured tabular logs with explicit object – event semantics and builds a heterogeneous graph, the high-level paradigm (dynamic graph + link prediction for log anomaly detection) is similar. The authors did not cite or discuss TempoLog. This omission significantly undermines the claimed novelty.

Qi et al., Beyond Window-Based Detection: A Graph-Centric Framework for Discrete Log Anomaly Detection, https://arxiv.org/abs/2501.12166,  Jan 2025.

- Baseline comparison is limited: This paper positions itself as a novel, general-purpose framework for tabular log debugging by contrasting primarily against a simple MLP (Multi-Layer Perceptron) and heavyweight RAG+LLM (Retrieval-Augmented Generation + Large Language Model) pipelines. However, it omits comparisons with a range of recent, highly relevant graph-based log anomaly detection methods — most notably TempoLog, GLAD-PAW, GLAD, LogGD, and OCDiGCN (see below the references). These recent methods all belong to the family of graph-based log analysis, sharing the common approach of modeling logs as dynamic or semantic-rich graphs. This significant omission could undermine the validity of the evaluation, making it difficult to determine the effectiveness of the proposed approach.

Y. Wan, Y. Liu, D. Wang, and Y. Wen, “Glad-paw: Graph-based log anomaly detection by position aware weighted graph attention network,” in Pacific-Asia conference on knowledge discovery and data mining, pp. 66–77, Springer, 2021.

Y. Xie, H. Zhang, and M. A. Babar, “Loggd: Detecting anomalies from system logs with graph neural networks,” in 2022 IEEE 22nd International conference on software quality, reliability and security (QRS), pp. 299–310, IEEE, 2022.

Z. Li, J. Shi, and M. Van Leeuwen, “Graph neural networks based log anomaly detection and explanation,” in Proceedings of the 2024 IEEE/ACM 46th International Conference on Software Engineering: Companion Proceedings, pp. 306–307, 2024.

Y. Li, Y. Liu, H. Wang, Z. Chen, W. Cheng, Y. Chen, W. Yu, H. Chen, and C. Liu, “Glad: Content-aware dynamic graphs for log anomaly detection,” in 2023 IEEE International Conference on Knowledge Graph (ICKG), pp. 9–18, IEEE, 2023.

- Concern regarding synthetic anomaly injection: The proposed method is evaluated using anomalies generated by randomly swapping existing objects or events within the log (Section 5.1). While this setup enables controlled experiments, it may not reflect realistic failure modes in real-world tabular logs. Such "swapped" entries often remain syntactically and semantically valid. In contrast, real systems typically encounter many anomaly scenarios such as semantic contradictions, temporal violations, out-of-distribution values, or structural inconsistencies — none of which can be captured by simple entity swapping. This synthetic anomaly model significantly weakens the validity of the evaluation and raises doubts about the method’s practical utility.

⦁ Dataset: The evaluation uses HDFS as the system log dataset. This dataset is relativly easier. The authors may consider other commonly-used system log datasets such as BGL, Thunderbird, and Spirit.

**Questions:**

- How is the proposed approach compared with recent graph-based log anomaly detection baselines?
- Do the experiments reflect real-world anomaly scenarios ?

---

> ### Author Response · Authors · 2025-11-19
>
> We thank the reviewer for the insightful comments and address the specific concerns below:
>
> [W1] Novelty and Comparison with TempoLog: We acknowledge that TempoLog (arXiv:2501.12166) is a recent and relevant work and will cite it in our revised draft. However, we disagree that the omission undermines our novelty, as GraphLogDebugger and TempoLog solve fundamentally different problems using distinct graph structures, despite sharing the high-level "dynamic graph" paradigm.
>
> **Heterogeneous Entity-Relation vs. Homogeneous Temporal Sequence**: In TempoLog, edges represent temporal adjacency, with the primary goal of modeling the sequential evolution of system states (i.e., "what event happens next?"). In GraphLogDebugger, we construct a heterogeneous graph where nodes are Objects (e.g., UserID, Paper Author, Stock Symbol) and Events (Log Content). Edges represent the interaction between an entity and an event (e.g., Author A wrote Paper B). Our goal is to model the structural consistency of entities within an event (i.e., "does this object belong in this event?").
>
> **Tabular vs. discrete Log**: TempoLog targets discrete system logs (unstructured text parsed into templates). It relies heavily on the sequence of templates. GraphLogDebugger targets tabular logs (structured data with explicit attributes). It leverages the rich, non-sequential relationships between columns (Objects/Features) and the event content.
>
> **Anomaly Definition**: TempoLog detects temporal anomalies (e.g., Event B following Event A is unexpected). GraphLogDebugger detects relational anomalies (e.g., User X performing Operation Y is unexpected, even if Operation Y is temporally valid).
>
> [W2] Baseline Comparisons: We appreciate the references to GLAD-PAW, GLAD, LogGD, and OCDiGCN. We omitted them primarily because our framework targets general tabular logs (including finance, academic, and system logs), whereas the cited methods are specialized exclusively for system log sequences.
>
> **Incompatibility with General Tabular Data**: Methods like LogGD and GLAD rely on extracting templates from repetitive system logs and modeling their sequential transitions. They cannot be easily applied to non-system domains like our Arxiv (academic papers) or Analyst (financial news) datasets, where "templates" are ill-defined or data is not strictly sequential.
>
> **Why MLP and RAG?** We compared our method against MLP and RAG because they are General-Purpose methods capable of handling the diverse tabular/text data we evaluated.
>
> [W3] Synthetic Anomaly Injection: We respectfully argue that our anomaly generation method (negative sampling via swapping) is both standard for link-prediction tasks and effective for detecting the "semantic contradictions" the reviewer mentions.
>
> **Swapping Creates Semantic Contradictions**: By swapping an object (e.g., Block 123) into a different event (e.g., Event B), we actively create a semantic contradiction where an object appears in a context where it does not belong. This directly mimics real-world errors such as misconfigurations, data corruption, or unauthorized access (e.g., a User accessing a resource they historically never touch).
>
> **Standard Practice**: In graph learning, determining if an edge (Object, Event) is valid is formulated as Link Prediction. Swapping entities to create non-existent edges is the standard negative sampling technique to train and evaluate such models.
>
> **Limitations in "Real" Anomalies**: Public tabular datasets with ground-truth "logic bugs" or "semantic inconsistencies" are extremely rare. Synthetic injection allows us to controllably evaluate the model's ability to learn valid entity-event correlations.
>
> [W4] Dataset Selection: We utilized HDFS as a representative system log dataset, but our primary focus is on the breadth of domains (Academic, Finance, Geology):
>
> **Redundancy of System Log Datasets**: Adding BGL, Thunderbird, and Spirit would heavily skew the evaluation towards system log parsing, which is the domain of the baselines mentioned in [W2], rather than the tabular debugging we propose.
>
> **Arxiv & Analyst**: These datasets showcase our model's unique ability to debug complex, non-system tabular data (e.g., detecting an author wrongly attributed to a paper), which is a key contribution distinct from traditional log anomaly detection.
> We will acknowledge this limitation and clarify that while HDFS is "easier" for sequence models, it effectively tests the entity-relation aspect our model targets.
>
> [Q1] Comparison with recent graph baselines: Our approach differs by modeling heterogeneous entity-event relations rather than homogeneous template sequences. We outperform general baselines (RAG/MLP) on diverse tabular tasks where sequence-based graph models are inapplicable.
>
> [Q2] Real-world scenarios: Yes, the "swapping" mimics structural/semantic inconsistencies (e.g., wrong entity in an event), which is a major class of real-world tabular errors.

---

### Note · Authors · 2026-01-05

I have read and agree with the venue's withdrawal policy on behalf of myself and my co-authors.